# Characterizing Healthy Dietary Practices in Japan: Insights from a 2024 Nationwide Survey and Cluster Analysis

**DOI:** 10.3390/nu16101412

**Published:** 2024-05-08

**Authors:** Shuhei Nomura, Akifumi Eguchi, Keiko Maruyama-Sakurai, Ruka Higashino, Daisuke Yoneoka, Takayuki Kawashima, Yuta Tanoue, Yumi Kawamura, Rauniyar Santosh Kumar, Takanori Fujita, Hiroaki Miyata

**Affiliations:** 1Department of Health Policy and Management, School of Medicine, Keio University, 35 Shinanomachi, Shinjuku-ku, Tokyo 160-8582, Japan; siero5335@gmail.com (A.E.); ksakurai@keio.jp (K.M.-S.); luca218r@keio.jp (R.H.); kawashima@c.titech.ac.jp (T.K.); amiyumiuni@gmail.com (Y.K.); rauniyar.sam@gmail.com (R.S.K.); taksen1979@gmail.com (T.F.); h-m@keio.jp (H.M.); 2Data for Social Transformation, 1-11-2 Osaki, Shinagawa-ku, Tokyo 141-0032, Japan; 3Department of Global Health Policy, Graduate School of Medicine, The University of Tokyo, 7-3-1 Hongo, Bunkyo-ku, Tokyo 113-0033, Japan; 4Center for Preventive Medical Sciences, Chiba University, 1-33, Yayoicho, Inage-ku, Chiba 263-8522, Japan; 5Tokyo Foundation for Policy Research, 3-2-1 Roppongi, Minato-ku, Tokyo 106-6234, Japan; 6Center for Surveillance, Immunization, and Epidemiologic Research, National Institute of Infectious Disease, 1-23-1 Toyama, Shinjuku-ku, Tokyo 162-0052, Japan; yoneoka@niid.go.jp; 7Department of Mathematical and Computing Science, Tokyo Institute of Technology, 2-12-1 Ookayama, Meguro-ku, Tokyo 152-8552, Japan; 8Faculty of Marine Technology, Tokyo University of Marine Science and Technology, 2-1-6, Etchujima, Koto-ku, Tokyo 135-8533, Japan; ytan001@kaiyodai.ac.jp; 9Ocean Policy Research Institute, Sasakawa Peace Foundation, 1-15-16 Toranomon, Minato-ku, Tokyo 105-8524, Japan

**Keywords:** dietary habits, dietary diversity, clustering, Japan

## Abstract

The increasing burden of lifestyle-related diseases highlights the need to address unhealthy dietary habits. This study aims to explore the latest dietary patterns in Japan following the COVID-19 pandemic, focusing on trends in health-promoting food choices. A web-based survey was conducted among 27,154 Japanese adults, selected via quota sampling to mirror national demographics. The study evaluated dietary diversity, measured through the Dietary Variety Score (Outcome 1), and the prioritization of nutritional and health considerations in food selection, assessed via a Likert scale (Outcome 2). Uniform Manifold Approximation and Projection (UMAP) and Ordering Points To Identify the Clustering Structure (OPTICS) algorithms were used to delineate patterns in health-centric food selections. OPTICS clustering revealed four distinct clusters for each outcome. Cluster 3, with a diverse diet, comprised older, predominantly female individuals with higher well-being and lower social isolation compared to Cluster 4, which lacked distinct dietary patterns. Cluster 3 also engaged more in snacking, treat foods, home cooking, and frozen meals. Similarly, a divide emerged between those prioritizing dietary considerations (Cluster C) and those indifferent to such aspects (Cluster D). The findings underscore the need for holistic post-COVID-19 public health initiatives addressing socioeconomic and cultural barriers to healthier dietary practices.

## 1. Introduction

Our diet serves not only as a source of pleasure and an integral part of culture in daily life but also as an essential behaviour for maintaining health. Unhealthy dietary habits are one of the primary causes of the increasing trend in lifestyle-related diseases (such as obesity, diabetes, heart diseases, and strokes, which are chronic conditions) that significantly impact our health. According to the latest Global Burden of Disease Study, unhealthy diets are responsible for 7.4% of the global health loss, measured in Disability-Adjusted Life Years (DALYs), and 6.6% of health loss in Japan as of 2019 [1].

Furthermore, the COVID-19 pandemic has led to the emergence of a “syndemic”, a term that describes the interaction between the increase in chronic disease burden and the pandemic, exacerbating people’s vulnerability to infectious diseases. The delay in public health measures to address preventable risk factors, such as unhealthy diets, has contributed to accelerating the pandemic [2]. The ongoing aging of populations worldwide, including in Japan, and the resulting increases in healthcare and social security costs, underscore the urgent need to build sustainable health systems capable of addressing chronic disease risk factors and aging societies to combat future health crises.

Diet plays a central role not only in physical nutrition but also in enhancing well-being and social relationships, enriching life in multiple dimensions. Systematic reviews have shown that enhancing the quality of dietary experiences, including the frequency of family meals and the diversity of consumed foods, can impact various aspects of health, including reducing smoking and drinking, preventing violent behaviour and suicide, and preventing mental health disorders [3,4,5,6]. Consequently, improving dietary habits and experiences has gained significant attention as an effective intervention for preventable health risks, both mentally and socially. While meta-analyses have confirmed the potential effectiveness of health intervention strategies such as taxing fructose beverages [7], improvements in the food environment through retail display signage and vending machine pricing adjustments [8,9], and enhancements in public dining facilities, including school cafeterias, workplace dining, restaurant menus, and vending machine product offerings [10,11], the majority of these studies are based on evidence from Western contexts, highlighting a gap in geographical universality, such as in Japan and Asia.

The perspectives and values regarding diet vary significantly across cultures, implying that Western findings may not be directly applicable to Asian nation [12], including Japan [13]. With increasing health consciousness and diversification of food values in society, the COVID-19 pandemic has further accelerated lifestyle changes and technological innovations in dietary experiences. Despite the essential role of evidence-based on Japan’s evolving dietary habits and experiences for national health improvement, research in this area is not yet sufficiently advanced. This study aims to generate evidence based on the latest dietary habits post-pandemic and experiences deeply rooted in Japanese culture. Specifically, our objective is to characterize the multifaceted aspects of dietary habits and experiences related to healthy food choices, utilizing dimensionality reduction and clustering techniques. Identifying clusters of healthy dietary choices and elucidating the unique characteristics inherent to these clusters will significantly contribute to understanding the dietary habits and experiences of Japan and other Asian countries. This understanding is crucial for improving public health through interventions against unhealthy dietary practices.

## 2. Materials and Methods

### 2.1. Study Participants

The participants of this study were registered members of a web survey company’s panel, Cross Marketing Inc. The panel included individuals over the age of 20 who were capable of responding to surveys in Japanese. Membership in the panel was voluntary, and as an incentive, respondents were awarded “points” based on the volume of surveys completed. These points could be utilized to purchase goods and services from partner companies. As of 2024, this survey company had access to approximately 5 million panel members domestically, representing a diverse array of demographic, socioeconomic, and geographic characteristics [14].

To ensure national representativeness, this study employed a quota sampling method based on age, gender, and regional population ratios derived from the 2020 Census [15], ultimately setting a fixed number of 27,905 participants. Participation was on a first-come, first-served basis, concluding once the predetermined target population for age, gender, and region was met. The survey commenced on 1 February 2024, achieving its target by 14 February 2024. Respondents were required to answer all questions to avoid missing values. To prevent disinterested or insincere responses, participants were asked to pledge to take the survey seriously before responding [16]. Masuda et al. (2019) found that participants who took the pledge were less likely to choose straight-lining (selecting ‘yes’ or ‘no’ across all items) or midpoint responses, indicating adherence to their initial commitment [16].

### 2.2. Measurement

The survey questionnaire comprised four sections: socioeconomic background information; life satisfaction (measured by well-being indicators) and social connections; habitual dietary preferences and tendencies, including healthy food choices (i.e., the two outcomes of the present study); and everyday meal experiences. Further details are described below. The questionnaire items were developed based on a thorough review of literature on similar topics, including studies conducted within the Japanese context [17,18,19,20,21,22,23,24,25]. This process was supervised by experts involved in the “Data for Social Transformation” academic platform, a collaboration among Japan’s government, industry, academia, and social sectors. Although our methodology did not include elements such as a pilot study, the survey design integrated expert opinions and was reflective of methods validated in prior research.

All questions were closed-ended, encompassing over 30 items in formats such as binary, yes/no, nominal, ordinal, and Likert scales. The content of the questions and response options are summarized in the resulting tables. The Likert scale regarding importance was treated as a continuous variable for simplicity, with the mean value being used as the unit of aggregation. Unless otherwise stated, respondents self-reported information at the time of the survey response.

### 2.3. Outcomes

This paper focuses on two primary outcomes. The first outcome (Outcome 1) is the Dietary Variety Score (DVS), a measure of dietary diversity developed by Kumagai et al. (2003) [26]. The DVS is determined through responses to a questionnaire inquiring about the frequency of consumption of ten specific food groups over the preceding seven days. Respondents choose from four options for each food group: ‘almost every day’; ‘every other day’; ‘1 to 2 times’; or ‘rarely eaten’. Scoring is binary, with ‘almost every day’ responses receiving one point and all other responses zero, thereby generating a binary variable for each food group. The food groups include seafood; soy and soy products; green and yellow vegetables; meats; eggs; fats and oils; seaweeds; tubers; fruits; and milk. Table 1 presents a detailed breakdown of these groups. The order of the food groups is randomized for each respondent, and answering all questions is mandatory. The DVS is utilized broadly in research exploring the link between dietary habits and health outcomes [22,23], as well as in practical applications for health promotion and preventive care [24,25].

The second outcome assessed in this study is an indicator frequently used to measure the level of importance regarding nutritional and health aspects in the selection of meals, ingredients, and groceries (Outcome 2). This measure employs a seven-point Likert scale, from ‘not at all important’ to ‘extremely important’, to assess responses to a question concerning the importance attributed to nutritional and health factors when selecting meals and food products. The evaluation covers eight specific dietary and nutritional aspects: reduction of salt; reduction of sugar; reduction of artificial additives; reduction of saturated fats; reduction of calories; increase of vitamins; increase of dietary fiber; and increase of unsaturated fats.

### 2.4. Additional Survey Items

#### 2.4.1. Survey Section I

In this study, we considered a range of sociodemographic variables including age, gender, body-mass index (BMI) calculated from self-reported height and weight in kilograms, both reported to one decimal place, place of residence by prefecture, educational level, type of occupation, household income in 2022, marital status, smoking status, frequency of alcohol consumption, current health status, usage of wearable devices or IoT (Internet of Things) appliances, utilization of social media platforms, and medical history.

#### 2.4.2. Survey Section II

Well-being was assessed using a method called the Cantril Ladder, developed by the Gallup World Poll, and this method has been utilized in the World Happiness Report [27]. This method involves asking participants how satisfied they are with their current life and their expectations for their life in five years, using a Likert scale ranging from 0 (completely dissatisfied) to 10 (completely satisfied). A binary indicator defines respondents with a current life rating of 7 or above and a future life expectation of 8 or above as having good well-being [27]. Additionally, social connections were evaluated using the Lubben Social Network Scale-6 (LSNS-6), developed by Lubben et al. (2003) [28,29]. The LSNS-6, focusing on emotional and instrumental support, comprises six questions divided equally between family and non-family networks, using a six-point scale to quantify the number of people in each network. Scores range from 0 to 30, with a score below 12 indicating social isolation.

#### 2.4.3. Survey Section III

The study explored dietary habits and preferences, focusing on the frequency of beverage consumption; the significance of diverse factors in the selection of foodstuffs and groceries; the importance of various aspects related to dining environments and settings outside the home; and the frequency and typical timing of food and drink consumption during “break times” or “snacking” between meals; and whether there are certain dining scenarios or circumstances (for example, days spent alone, special occasions, or during business trips) when individuals might choose foods they usually would not select or find difficult to choose, often termed as “reward foods”.

#### 2.4.4. Survey Section IV

Regarding daily meal experiences, the study examines whether individuals prepare their meals themselves; the number of meals consumed alone in the past seven days for each mealtime (breakfast, lunch, and dinner); the conversational context experienced during meals for each mealtime; satisfaction with meals for each mealtime; perceived time availability during meals for each mealtime; the frequency of various types of meals; the frequency of different methods for purchasing foodstuffs and groceries; the most commonly used modes of transportation for shopping for food and groceries; the frequency of participation in social situations or events centered around meals; and the use of technology or applications related to meals, food, and groceries.

### 2.5. Data Analysis

In this study, we conducted separate analyses for each of two outcomes to characterise the overarching features of participants’ healthy dietary choices. We utilized Uniform Manifold Approximation and Projection (UMAP) and Ordering Points To Identify the Clustering Structure (OPTICS) algorithms for this analysis. UMAP, a dimensionality reduction technique, enables the identification of the overall structure of data [30], as detailed in the Appendix A. For each outcome, we reduced ten binary variables and eight 7-point Likert scale variables to a two-dimensional space for visual inspection. Subsequently, the OPTICS algorithm, a developed density-based clustering which does not determine the number of clusteres in advance [31], was adopted to identify meaningful clusters of individuals within the dimensionality-reduced space. Unlike many known clustering algorithms that require parameter values to be input before analysis, which is challenging and significantly impacts results, OPTICS necessitates minimal parameters: MinPts and epsilon. In UMAP, a fixed number of nearest neighbours (set to 50 for both outcome) was employed using Hamming distance for binary and Likert scale variables. For the OPTICS algorithm, MinPts was set to the number of 5% of the data points (default is 5 data points), and epsilon were set to 7.5 and 10 for binary and Likert scale variables, respectively. To see the plots of clustering, we set clustering reachability distance thresholds to 7.0 and 7.5 for the two outcomes, respectively, and employed simple heuristics for visually determining thresholds, as long as the distance values were not too small (Appendix A), following OPTICS usage guidelines [31,32]. Data analysis was performed using R version 4.3.0 and the packages ‘uwot’ and ‘dbscan’.

After clusters were identified and respondents assigned to each cluster, we aggregated the response results for each of the ten or eight survey variables associated with the two outcomes within each cluster. We then focused on the differences in the response results between clusters as a comprehensive characteristic to evaluate the significant features of each cluster.

Finally, we aggregated data for all four sections of survey items within each cluster to compare differences between clusters, aiming to understand their distinct characteristics comprehensively. The outcomes of the respondents, such as mean values and percentages, were compared among clusters utilizing ANOVA tests, followed by post-hoc analyses with Dunnett’s test, or chi-squared tests as appropriate. The selection of the reference cluster for comparisons was made with a pragmatic approach, prioritizing comparisons that offer policy-relevant insights. A *p*-value below 0.05 was deemed statistically significant. To control for type I error inflation due to multiple comparisons, the Bonferroni correction was employed where necessary to adjust for multiple testing.

## 3. Results

In the survey of 27,905 respondents, analyses were conducted excluding 751 individuals (2.7%) who provided responses with improbable body weight or height, indicating potential data entry errors. This determination was based on values that deviated by more than 1.5 times the interquartile range (IQR) from the first and third quartiles of the BMI median, resulting in 27,154 valid responses (97.3%).

The sociodemographic characteristics of the respondents are presented in Table 2. The average age at the time of the survey was 53.6 years, with a standard deviation of 16.6. The gender distribution slightly favored males (51.6%). Among the respondents, 45.6% had received post-secondary education, 60.3% were married, and 56.4% had a history of some pre-existing condition.

Using the OPTICS clustering algorithm, we detected four subpopulations (clusters) for each of two outcomes. Figure 1(a1,a2) depict the distribution of clusters for Outcomes 1 and 2, respectively. Figure 1(b1,b2) visualize the clusters identified for Outcomes 1 and 2 through a two-dimensional reduction representation using UMAP.

Table 3 and Table 4 display cluster-specific responses to survey questions, with ten questions related to Outcome 1 and eight to Outcome 2, respectively. For Outcome 1, the clusters are described as follows: Cluster 1 includes individuals who consume meat or fats/oils daily (3.1%, *n* = 845); Cluster 2 consists of those who drink milk daily (6.0%, *n* = 1641); Cluster 3 is characterized by respondents who maintain a well-rounded daily diet (67.2%, *n* = 18,244); and Cluster 4 includes individuals without a specific daily dietary pattern (23.7%, *n* = 6424).

For Outcome 2, the clusters are defined as: Cluster A is characterized by respondents with extreme response tendencies, who exclusively selected the highest and lowest scores on the Likert scale, indicating polarized views on the importance of nutrients (5.5%, *n* = 1483). Cluster B consists of individuals who generally perceive all nutrients as unimportant, predominantly selecting the lower end of the scale to express their disregard for nutrient importance (9.5%, *n* = 2573). Cluster C includes those who deem all nutrients as generally important, likely choosing higher scores on the scale to reflect their valuation of nutrient importance (64.1%, *n* = 17,401). Finally, Cluster D is made up of respondents who exhibit a neutral stance towards nutrient importance, frequently opting for the middle option on the scale, indicating an indifference or lack of conscious consideration towards the importance of nutrients (21.0%, *n* = 5697).

Our study uncovers notable differences between clusters, especially when comparing those with varied daily diets to those without specific dietary patterns. Individuals in Cluster 3, who have a varied daily diet, compared to the reference group, Cluster 4, were found to be on average older, predominantly female, more educated, more likely to be married, less prone to smoking, and had a higher prevalence of medical conditions (Appendix A). Cluster 3 individuals exhibited a higher well-being percentage (24.3% vs. 14.1%) and a lower rate of social isolation (59.9% vs. 73.6%). They valued a broad spectrum of food selection and dining preferences more highly, excluding aspects related to digital and modern ordering systems. The engagement in snacking was more common in Cluster 3 (41.7% vs. 24.1% snacking at least once daily), as was the consumption of ‘treat foods’ under specific conditions (31.9% vs. 24.7%). This cluster also reported a greater allocation of time for meals and a higher frequency of home cooking and frozen meal consumption (42.1% vs. 37.8% consuming frozen meals at least weekly). Individuals in Cluster 3 were less likely to engage in solitary dining or meals without conversation. Furthermore, Cluster 3 displayed a notable integration of dining with technology and applications, especially in terms of recipe suggestion app usage, which stood at 24.0% compared to just 12.3% in Cluster 4. Following the COVID-19 pandemic, there was also an increased awareness regarding dietary choices in Cluster 3. This heightened awareness led to a significant uptick in the stockpiling of durable food items and groceries, such as canned and frozen foods (16.6% in Cluster 3 versus 9.4% in Cluster 4), and a growing interest in recipes that reduce cooking time and simplify meal preparation (10.2% compared to 6.6%).

When comparing Cluster C, which emphasizes the importance of diet, to Cluster D, which shows a lack of concern for dietary habits, similar patterns emerged (Appendix A). We performed a chi-squared test using a contingency table for clusters 1–4 and clusters A–D, revealing a significant association between these clusters (*p* < 0.001). Members of Cluster C exhibited behaviors and preferences that mirrored those in Cluster 3, with 70.3% of Cluster 3’s members belonging to Cluster C; similarly, 73.4% of Cluster C’s members were also part of Cluster 3. These shared characteristics include better well-being, reduced social isolation, and a predilection for snacking, treat food consumption, home cooking, utilizing frozen meals, integrating dining with technology and applications, and a heightened awareness of dietary choices following the pandemic. Additionally, within Cluster C, there was a notably lower occurrence of individuals engaging in solitary dining or meals without conversation. For a comprehensive understanding of these patterns and their implications, please refer to the Appendix A, which provide in-depth analyses and data supporting these findings.

Cluster 1, comprised of individuals who consume meat or fats/oils daily, exhibited tendencies that were partially similar to those observed in Cluster 4. Conversely, Cluster 2, characterized by daily milk consumption, showed similarities to Cluster 3. Regarding well-being and social isolation, Cluster 1 (17.3% and 69.6%, respectively) demonstrated no significant differences when compared to Cluster 4 (14.1% and 73.6%, respectively). In contrast, Cluster 2 (17.7% and 66.1%, respectively) exhibited significant differences from Cluster 4. Despite these findings, the dietary habits and experiences relative to Cluster 4 were generally varied, with both Clusters 1 and 2 showing fewer significant variables compared to Cluster 3. The interpretation of responses from Cluster A, which displayed extreme views on the importance of nutrition, continues to be challenging. On the other hand, Cluster B, which generally deems all nutrients as unimportant, had dietary habits and experiences that were closely aligned with those of Cluster D, with no significant differences in well-being and social isolation (16.8% and 72.3% vs. 16.6% and 71.3%, respectively). Nonetheless, variations in dietary habits and experiences were still observed across the clusters. Further details are available in Appendix A.

## 4. Discussion

The contrast between Cluster 3, characterized by a diet rich in variety, and Cluster 4, which lacks a specific dietary pattern, underscores the potential role of a diverse diet in improving health outcomes. The demographic profile of Cluster 3—older age, predominantly female, higher education levels, and a higher likelihood of being married—indicates that dietary choices are deeply linked to lifestyle and socioeconomic status. This insight supports the notion that interventions aimed at dietary improvement should be multifaceted, addressing not only the nutritional value of food but also the socioeconomic barriers to healthy eating. This viewpoint aligns with findings from other studies, such as those by Scander (2021), which underscore the importance of social context of diet in assessing nutritional health [33], resonating with our research that highlights the multifaceted nature of food choices. Scander’s findings suggest that interventions targeting dietary improvements must consider the broader socioeconomic and cultural factors that influence when and with whom individuals eat [33,34].

Furthermore, the notable sense of well-being and lower rates of social isolation observed in Cluster 3 suggest a link between diet quality, mental health, and social well-being, corroborating existing literature. For example, Sakurai’s (2021) study, which explored the association between dining alone, weak social networks, and depression among older Japanese individuals residing alone, highlights the significant impact of social connections on food choices and mental health [20]. Yamaguchi et al. (2020) also studied the older adults in Japan and found that a lack of social participation was inversely related to the frequency of vegetable and fruit consumption [35]. These findings indicate that individuals experiencing a lack of social involvement were less inclined towards adopting a healthy diet. This relationship may be partially explained by the mechanism where social networks and communal activities, particularly in cultures valuing group harmony like Japan, enhance motivation for healthy behaviours through mutual support and shared values [35,36]. These insights complement our findings by underscoring the importance of addressing social isolation and promoting communal dining experiences as part of dietary interventions, particularly in aging societies like Japan.

Investigations into the relationship between communal eating and dietary intake patterns provide insights into how shared meal opportunities can promote healthier eating behaviors. Research by Dallacker (2018) and Verhage (2018) offers evidence of a positive association between family meals and nutritional health across diverse demographic groups, including children and adolescents [4,6]. These findings suggest that communal eating facilitates access to nutritious foods, encourages balanced eating habits, and serves as an anchor for establishing routine and structure around meal times. The sociocultural significance of dining together extends to the realm of mental well-being, with qualitative research by Bascuñan-Wiley (2022) and a systematic review by Harrison (2015) indicating that shared meal times strengthen social cohesion, foster a sense of belonging, and enhance overall life satisfaction [5,37]. This is particularly relevant in contexts where social bonds are under pressure, such as during the COVID-19 pandemic or among populations experiencing rapid cultural transformation.

The differences in engagement with food-related technology among clusters highlight the evolving landscape of food consumption. The pandemic has particularly accelerated the adoption of technology in dietary choices, especially regarding smartphone apps. This shift presents both challenges and opportunities for public health strategies, suggesting that interventions leveraging technology to promote healthy eating should be implemented, while also considering the digital divide. Our research, focusing on the Japanese context, synergizes with international findings by Wang (2023) and Spence (2019), which explore the impact of digital communal eating and technological interventions on food choices and mental health [38,39]. These studies suggest that technology-mediated social dining experiences can mitigate loneliness and foster social connections, thereby enhancing both physical and mental health [38,39].

The contrast between Cluster C, which prioritizes nutrition, and Cluster D, indifferent to the quality of their diet, highlights the significant impact of education and awareness on shaping dietary habits. Increasing knowledge about nutrition and enhancing the accessibility and affordability of healthy food options are crucial in guiding individuals towards healthier eating habits. This approach is particularly relevant in addressing the ‘syndemic’ of lifestyle diseases and infectious diseases, as a healthier population is likely to be more resilient to health crises [40,41]. Studies have shown that nutrition education plays a vital role in improving dietary quality among different age groups, including children and pregnant women, emphasizing the positive outcomes of educational interventions on dietary habits [42,43]. Additionally, research has explored the relationship between educational attainment and nutrition, highlighting the interplay between education levels and diet composition, underscoring the importance of educational initiatives in promoting healthier food choices [44].

Particularly within clusters characterized by a diverse daily diet and a high valuation of dietary health, the consumption of frozen foods emerged as a notable behavior. This finding challenges the traditional perception that frozen foods are inferior in health quality or less desirable compared to fresh foods [45,46]. Despite initial skepticism, frozen foods have become an integral part of the daily diet in many countries, with their consumption figures and market turnover continuing to increase [46]. The growing acceptance and reliance on frozen foods can be attributed to several factors. Firstly, technological advancements have significantly improved the nutritional quality, variety, and taste of frozen foods, making them a viable alternative to fresh meals [47]. This development holds particular significance in Japan, characterized by its aging demographic and the often time-constrained urban lifestyle, where the simplicity of preparation and storage of these foods can substantially ease meal management for both the older population and busy professionals. Furthermore, the COVID-19 pandemic has led to a notable increase in individuals stockpiling long-lasting food items, such as frozen food, in these clusters.

Our research, rooted in the Japanese context, unravels the multifaceted nature of dietary choices, offering a lens through which to view the universal challenge of improving public health through better nutrition. The identification of distinct dietary clusters advocates for tailored approaches in dietary interventions. Such strategies should aim not only to educate and provide information but also to respect and integrate cultural values and norms into public health initiatives.

### Limitations

The present study is subject to several limitations. The principle limitation of this study stems from its exclusive focus on behaviors and attitudes related to the choices of a healthy diet, without conducting a direct evaluation of the resulting health outcomes. Behaviors and attitudes towards nutritional choices act as significant indicators for the potential enhancement of health; however, essentially, they do not guarantee visible health benefits, making this distinction extremely important [48].

Second, self-selection bias may affect the representativeness of the survey participants. However, it is important to highlight that despite potential biases, the study effectively uses incentives in the form of points (usable for purchasing goods) by the survey company. These incentives are designed to motivate responses from individuals otherwise disinterested in the survey, thereby potentially mitigating some of the common limitations associated with self-reported data. Furthermore, participants were recruited using a stratified sampling method based on age, gender, and regional population ratios as specified by the 2020 National Census, making the survey reasonably representative of the Japanese population. However, the educational level of respondents is comparatively higher than that identified in the census. Specifically, 45.6% of respondents aged 20 and above have attended university or possess a higher education level, significantly exceeding the percentage of the total population aged 15 and above with the same educational background, as reported by the 2020 National Census [15].

Third, the study’s reliance on self-reported data, particularly for sensitive information such as dietary habits, health status, and socioeconomic background, may lead to reporting bias. Participants might have overreported socially desirable behaviors or underreported socially undesirable ones, affecting the accuracy of the data collected.

Fourth, the timing of our survey could also affect the findings. It was conducted over two weeks in early February 2024, utilizing a 7-day food frequency questionnaire. While many food items are available year-round, particularly frozen foods, seasonal variations still exist and could influence respondents’ habitual dietary preferences, food choices, and daily meal experiences. This potential seasonal bias is an important factor to consider in interpreting our results.

Fifth, the use of dimensionality reduction techniques (UMAP) and clustering analysis (OPTICS) allows for sophisticated data exploration, but these methods also have limitations. The interpretation of clusters and the dimensionality-reduced space can be subjective, and the outcomes might not capture all the nuances of the dietary habits and health aspects explored. The chosen parameters for these analyses, while based on heuristics and guidelines, could still influence the findings and their interpretation.

Sixth, the study’s cross-sectional design limits the ability to infer causality from the associations observed between dietary habits, health aspects, and other variables. Longitudinal data would be required to establish causal relationships and to understand the directionality of these associations.

Seventh, while the survey methodology employed in this study draws on expert opinions and a comprehensive literature review, it is important to note the limitations associated with the absence of a pilot study and repeated measurements from the same respondents. These conditions are typically required for certain statistical validations such as the calculation of reliability coefficients. Consequently, the interpretations of the survey results should be considered with these methodological constraints in mind.

Finally, as the survey was conducted through a web panel, there may be limitations related to the accessibility of the technology required to participate. This could exclude individuals with limited internet access or technological proficiency, which might affect the diversity of the sample and the representativeness of the study.

## 5. Conclusions

In this concise study, we evaluated the dietary habits of 27,154 Japanese participants, revealing distinct dietary behavior clusters. This research underlines the importance of multifaceted interventions that go beyond nutritional education to address the socioeconomic and sociocultural barriers to healthy eating. The integration of communal dining experiences and the leveraging of technology in dietary strategies emerge as key elements in fostering healthier eating behaviors and enhancing social well-being. Furthermore, our research confronts traditional beliefs regarding frozen foods, indicating that technological progress has rendered them an integral part of a nutritious diet, particularly within scenarios such as Japan’s aging society and the time-pressured urban lifestyles. Our findings advocate for tailored public health strategies that are sensitive to sociocultural values and the dynamic nature of food consumption, emphasizing the need for comprehensive approaches in promoting dietary health and combating lifestyle diseases.

## Figures and Tables

**Figure 1 nutrients-16-01412-f001:**
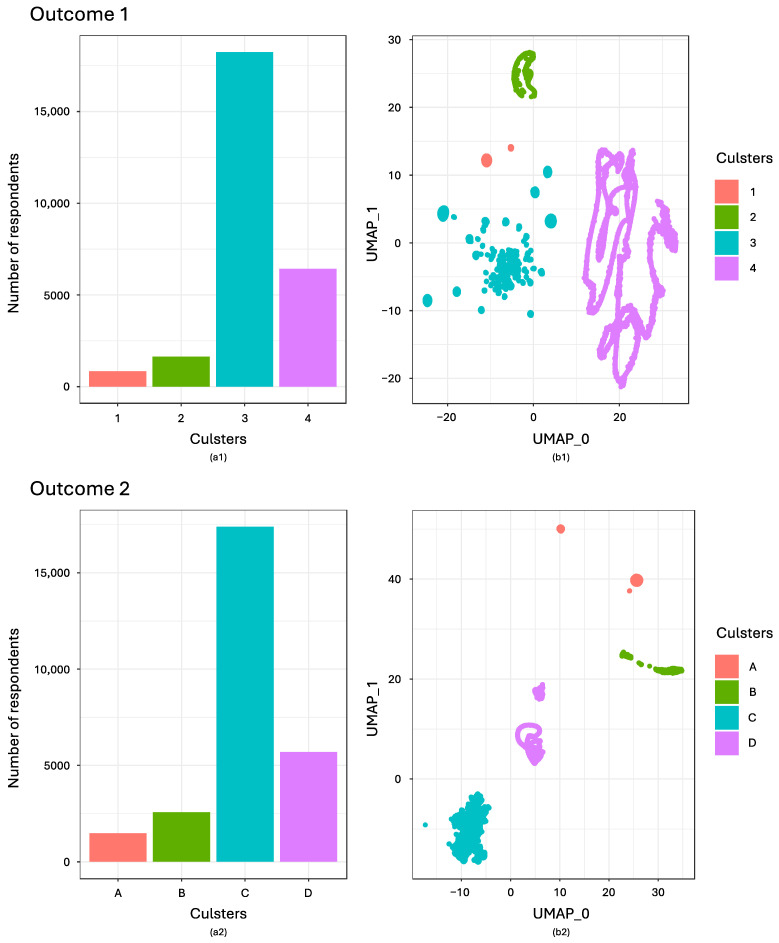
(**a**) Distribution of clusters detected by OPTICS on the two-dimensional reduced representation of the data; (**b**) UMAP clusters for two-dimensional reduced representation of the data annotated by the OPTICS generated clusters. Outcome 1 refers to the Dietary Variety Score (DVS), which assesses dietary diversity based on responses to a questionnaire with ten items. These items ask about the frequency of consumption of ten specific food groups—seafood, soy products, green and yellow vegetables, meats, eggs, fats and oils, seaweeds, tubers, fruits, and milk—over the past week. The scoring system is binary: daily consumption is awarded one point, while all other frequencies score zero, resulting in a binary score for each of the ten food groups. Clusters 1–4 in Outcome 1 are as described in Table 3. Outcome 2 refers to an indicator that measures the level of importance regarding nutritional and health aspects through a questionnaire comprising eight items. This measurement uses a seven-point Likert scale, ranging from ‘not at all important’ to ‘extremely important’, to evaluate the importance placed on nutritional and health factors when selecting meals and food products. The eight dietary and nutritional aspects covered include the reduction of salt, reduction of sugar, reduction of artificial additives, reduction of saturated fats, reduction of calories, increase of vitamins, increase of dietary fiber, and increase of unsaturated fats. Clusters A–D in Outcome 2 are as described in Table 4.

**Table 1 nutrients-16-01412-t001:** Detailed descriptions of food groups for the Dietary Variety Score.

Food Group	Description
Seafood	Includes all fish and shellfish, regardless of whether they are fresh or processed.
Soy and soy products	Covers foods made from soybeans, such as tofu and natto.
Green and yellow vegetables	Includes vegetables with rich colors, such as carrots, spinach, pumpkins, and tomatoes.
Meats	Includes all types of meat, both fresh and processed.
Eggs	Consists of eggs from chickens, quails, and other birds, excluding fish eggs.
Fats and oils	Includes dishes cooked with oil, such as stir-fries, tempura, and fried foods, as well as spreads like butter.
Seaweeds	Encompasses both fresh and dried varieties.
Tubers	Includes distinct category plants like potatoes and sweet potatoes.
Fruits	Includes all types, whether fresh or canned, but excludes tomatoes, which are classified as vegetables.
Milk	Refers specifically to cow’s milk and does not include flavored milks such as coffee milk or fruit milk.

The descriptions of each group are excerpted from Kumagai et al. (2003) [26].

**Table 2 nutrients-16-01412-t002:** Sociodemographic characteristics of the respondents.

Variables	Number of Respondents (%)
Age, mean (standard deviation)	53.6 (16.61)
Gender (SA)	
Female	13,003 (47.89)
Male	13,997 (51.55)
Other	154 (0.57)
Body-mass index (BMI), mean (standard deviation)	21.89 (3.17)
Residence (SA)	
Hokkaido	1079 (3.97)
Tohoku	1919 (7.07)
Kanto	9483 (34.92)
Chubu	4717 (17.37)
Kinki	4596 (16.93)
Chugoku	1631 (6.01)
Shikoku	797 (2.94)
Kyushu	2932 (10.8)
Educational Background (SA)	
Junior high school graduate	731 (2.69)
High school/technical college graduate or enrolled	8981 (33.07)
Junior college/vocational school graduate or enrolled	5057 (18.62)
University graduate or enrolled	11,206 (41.27)
Graduate school completed or enrolled	1179 (4.34)
Occupation (SA)	
Managerial occupation	1672 (6.16)
Professional or technical occupation	3020 (11.12)
Clerical worker	3504 (12.90)
Sales worker	1298 (4.78)
Service worker	2798 (10.30)
Security worker	223 (0.82)
Agriculture, forestry, and fisheries worker	181 (0.67)
Production process worker	1168 (4.30)
Transport and machinery operation worker	307 (1.13)
Construction and mining worker	327 (1.20)
Material moving, cleaning, packing, etc. worker	573 (2.11)
Student	435 (1.60)
Full-time homemaker	4603 (16.95)
Other (including unemployed, retired)	7045 (25.94)
Annual Income (SA)	
Less than 2 million yen/about Less than $15,000	4592 (16.91)
2 to under 4 million yen/about $15,400–$31,000	7931 (29.21)
4 to under 6 million yen/about $30,800–$46,000	6041 (22.25)
6 to under 8 million yen/about $46,200–$62,000	3851 (14.18)
8 to under 10 million yen/about $61,500–$77,000	2245 (8.27)
10 to under 20 million yen/about $76,900–$154,000	2058 (7.58)
Over 20 million yen/Over $154,000	436 (1.61)
Marital Status (SA)	
Married (including common-law marriage)	16,370 (60.29)
Single (no partner)	6462 (23.80)
Single (with a partner)	1548 (5.70)
Widowed	978 (3.60)
Divorced	1796 (6.61)
Smoking (SA)	
Smokes daily	4919 (18.12)
Smokes occasionally	454 (1.67)
Used to smoke but has not smoked for over a month	4867 (17.92)
Does not smoke	16,914 (62.29)
Drinking (SA)	
Daily	4495 (16.55)
5–6 days per week	1797 (6.62)
3–4 days per week	1744 (6.42)
1–2 days per week	3075 (11.32)
1–3 days per month	2574 (9.48)
Rarely drinks	4296 (15.82)
Stopped drinking	920 (3.39)
Does not drink (cannot drink)	8253 (30.39)
Health Condition (SA)	
Good	4145 (15.26)
Fairly good	7454 (27.45)
Average	11,071 (40.77)
Not very good	3593 (13.23)
Poor	891 (3.28)
Frequency of Device Use	
Wearable devices (SA)	
Almost every day	1879 (6.92)
2–5 days per week	475 (1.75)
About once a week or less	474 (1.75)
Do not use	24,326 (89.59)
IOT appliances (SA)	
Almost every day	833 (3.07)
2–5 days per week	402 (1.48)
About once a week or less	513 (1.89)
Do not use	25,406 (93.56)
Frequency of Social Media Use	
Facebook (SA)	
Almost every day	2451 (9.03)
2–5 days per week	1374 (5.06)
About once a week or less	2905 (10.70)
Do not use	20,424 (75.22)
X/Twitter (SA)	
Almost every day	6278 (23.12)
2–5 days per week	2049 (7.55)
About once a week or less	2444 (9.00)
Do not use	16,383 (60.33)
LINE (SA)	
Almost every day	14,694 (54.11)
2–5 days per week	4109 (15.13)
About once a week or less	2847 (10.48)
Do not use	5504 (20.27)
Instagram (SA)	
Almost every day	6091 (22.43)
2–5 days per week	1843 (6.79)
About once a week or less	2342 (8.62)
Do not use	16,878 (62.16)
Youtube (SA)	
Almost every day	10,128 (37.30)
2–5 days per week	4648 (17.12)
About once a week or less	5159 (19.00)
Do not use	7219 (26.59)
Tiktok (SA)	
Almost every day	2526 (9.30)
2–5 days per week	953 (3.51)
About once a week or less	1394 (5.13)
Do not use	22,281 (82.05)
Medical History	
Hypertension	5153 (18.98)
Diabetes	1687 (6.21)
Dyslipidemia (hyperlipidemia)	2512 (9.25)
Pneumonia/Bronchitis	960 (3.54)
Asthma	1519 (5.59)
Atopic dermatitis	1386 (5.10)
Allergic rhinitis	2511 (9.25)
Periodontal disease	3033 (11.17)
Dental caries (cavities)	4986 (18.36)
Cataract	1882 (6.93)
Angina/Myocardial infarction	594 (2.19)
Stroke (cerebral infarction, cerebral hemorrhage)	337 (1.24)
COPD (Chronic Obstructive Pulmonary Disease)	100 (0.37)
Chronic kidney disease	203 (0.75)
Chronic hepatitis/Cirrhosis	153 (0.56)
Immunodeficiency or immune function decline (including those on steroids, biologics, immunosuppressants)	271 (1.00)
Cancer/Malignant tumor	1484 (5.47)
Chronic pain (e.g., persistent back pain, headache for over three months)	1029 (3.79)
Depression	1232 (4.54)
Mental illness other than depression	996 (3.67)
None apply	11,839 (43.60)

SA refers to single-answer questions.

**Table 3 nutrients-16-01412-t003:** Cluster specific response for Outcomes 1.

	Cluster 1	Cluster 2	Cluster 3	Cluster 4
Number of Respondents (%)	*n* = 845, 3.11%	*n* = 1641, 6.03%	*n* = 18,244, 67.19%	*n* = 6424, 23.66%
Dietary Variety Score				
Seafood	32 (3.8)	21 (1.3)	2666 (14.6)	0 (0.0)
Soy products	0 (0.0)	0 (0.0)	8200 (44.9)	0 (0.0)
Green and yellow vegetables	1 (0.1)	0 (0.0)	10,694 (58.6)	0 (0.0)
Meats	601 (71.1)	0 (0.0)	6368 (34.9)	0 (0.0)
Eggs	0 (0.0)	0 (0.0)	8722 (47.8)	0 (0.0)
Fats and oils	241 (28.5)	0 (0.0)	6058 (33.2)	0 (0.0)
Seaweeds	34 (4.0)	0 (0.0)	3772 (20.7)	0 (0.0)
Tubers	7 (0.8)	18 (1.1)	1806 (9.9)	29 (0.5)
Fruits	0 (0.0)	0 (0.0)	7835 (42.9)	0 (0.0)
Milk	241 (28.5)	1641 (100.0)	8455 (46.3)	0 (0.0)

Outcome 1 refers to the Dietary Variety Score (DVS), which assesses dietary diversity based on responses to a questionnaire with ten items. These items ask about the frequency of consumption of ten specific food groups—seafood, soy products, green and yellow vegetables, meats, eggs, fats and oils, seaweeds, tubers, fruits, and milk—over the past week. The scoring system is binary: daily consumption is awarded one point, while all other frequencies score zero, resulting in a binary score for each of the ten food groups. Cluster 1 includes individuals who consume meat or fats/oils daily; Cluster 2 consists of those who drink milk daily; Cluster 3 is characterized by respondents who maintain a well-rounded daily diet; and Cluster 4 includes individuals without a specific daily dietary pattern.

**Table 4 nutrients-16-01412-t004:** Cluster specific response for Outcomes 2.

	Cluster A	Cluster B	Cluster C	Cluster D
Number of Respondents (%)	*n* = 1483, 5.46%	*n* = 2573, 9.48%	*n* = 17,401, 64.08%	*n* = 5697, 20.98%
Importance of considering nutritional and health aspects when choosing foods.				
Reduction of salt				
Not at all important	813 (54.8)	149 (5.8)	81 (0.5)	141 (2.5)
Minimally important	43 (2.9)	1054 (41.0)	157 (0.9)	318 (5.6)
Slightly important	8 (0.5)	929 (36.1)	376 (2.2)	704 (12.4)
Neither important nor unimportant	2 (0.1)	260 (10.1)	3309 (19.0)	3843 (67.5)
Somewhat important	14 (0.9)	143 (5.6)	8146 (46.8)	532 (9.3)
Quite important	33 (2.2)	30 (1.2)	3937 (22.6)	118 (2.1)
Extremely important	570 (38.4)	8 (0.3)	1395 (8.0)	41 (0.7)
Reduction of sugar				
Not at all important	804 (54.2)	197 (7.7)	19 (0.1)	226 (4.0)
Minimally important	50 (3.4)	1035 (40.2)	123 (0.7)	477 (8.4)
Slightly important	5 (0.3)	906 (35.2)	530 (3.0)	808 (14.2)
Neither important nor unimportant	7 (0.5)	272 (10.6)	4715 (27.1)	3802 (66.7)
Somewhat important	18 (1.2)	136 (5.3)	8068 (46.4)	318 (5.6)
Quite important	48 (3.2)	14 (0.5)	2991 (17.2)	39 (0.7)
Extremely important	551 (37.2)	13 (0.5)	955 (5.5)	27 (0.5)
Reduction of artificial additives				
Not at all important	811 (54.7)	271 (10.5)	105 (0.6)	255 (4.5)
Minimally important	24 (1.6)	1078 (41.9)	257 (1.5)	439 (7.7)
Slightly important	11 (0.7)	871 (33.9)	718 (4.1)	697 (12.2)
Neither important nor unimportant	14 (0.9)	247 (9.6)	5544 (31.9)	3847 (67.5)
Somewhat important	14 (0.9)	78 (3.0)	6727 (38.7)	328 (5.8)
Quite important	72 (4.9)	18 (0.7)	2861 (16.4)	84 (1.5)
Extremely important	537 (36.2)	10 (0.4)	1189 (6.8)	47 (0.8)
Reduction of saturated fats				
Not at all important	837 (56.4)	227 (8.8)	32 (0.2)	211 (3.7)
Minimally important	26 (1.8)	1112 (43.2)	147 (0.8)	488 (8.6)
Slightly important	1 (0.1)	942 (36.6)	501 (2.9)	851 (14.9)
Neither important nor unimportant	3 (0.2)	211 (8.2)	5653 (32.5)	3825 (67.1)
Somewhat important	21 (1.4)	70 (2.7)	7646 (43.9)	287 (5.0)
Quite important	66 (4.5)	8 (0.3)	2741 (15.8)	27 (0.5)
Extremely important	529 (35.7)	3 (0.1)	681 (3.9)	8 (0.1)
Reduction of calories				
Not at all important	808 (54.5)	215 (8.4)	178 (1.0)	296 (5.2)
Minimally important	34 (2.3)	1019 (39.6)	445 (2.6)	493 (8.7)
Slightly important	9 (0.6)	884 (34.4)	791 (4.5)	725 (12.7)
Neither important nor unimportant	25 (1.7)	293 (11.4)	5005 (28.8)	3734 (65.5)
Somewhat important	51 (3.4)	134 (5.2)	7634 (43.9)	352 (6.2)
Quite important	79 (5.3)	20 (0.8)	2666 (15.3)	69 (1.2)
Extremely important	477 (32.2)	8 (0.3)	682 (3.9)	28 (0.5)
Increase of vitamins				
Not at all important	782 (52.7)	224 (8.7)	14 (0.1)	14 (0.2)
Minimally important	41 (2.8)	1014 (39.4)	55 (0.3)	63 (1.1)
Slightly important	6 (0.4)	924 (35.9)	278 (1.6)	319 (5.6)
Neither important nor unimportant	10 (0.7)	281 (10.9)	3592 (20.6)	3994 (70.1)
Somewhat important	28 (1.9)	117 (4.5)	8307 (47.7)	998 (17.5)
Quite important	47 (3.2)	9 (0.3)	3966 (22.8)	239 (4.2)
Extremely important	569 (38.4)	4 (0.2)	1189 (6.8)	70 (1.2)
Increase of dietary fiber				
Not at all important	798 (53.8)	198 (7.7)	35 (0.2)	15 (0.3)
Minimally important	35 (2.4)	1016 (39.5)	102 (0.6)	44 (0.8)
Slightly important	8 (0.5)	1034 (40.2)	308 (1.8)	326 (5.7)
Neither important nor unimportant	6 (0.4)	221 (8.6)	2849 (16.4)	3933 (69.0)
Somewhat important	12 (0.8)	78 (3.0)	8183 (47.0)	1006 (17.7)
Quite important	35 (2.4)	19 (0.7)	4454 (25.6)	288 (5.1)
Extremely important	589 (39.7)	7 (0.3)	1470 (8.4)	85 (1.5)
Increase of unsaturated fats				
Not at all important	839 (56.6)	272 (10.6)	54 (0.3)	223 (3.9)
Minimally important	24 (1.6)	1122 (43.6)	173 (1.0)	448 (7.9)
Slightly important	0 (0.0)	906 (35.2)	560 (3.2)	732 (12.8)
Neither important nor unimportant	3 (0.2)	233 (9.1)	5747 (33.0)	3850 (67.6)
Somewhat important	6 (0.4)	34 (1.3)	7251 (41.7)	360 (6.3)
Quite important	62 (4.2)	5 (0.2)	2822 (16.2)	66 (1.2)
Extremely important	549 (37.0)	1 (0.0)	794 (4.6)	18 (0.3)

Outcome 2 refers to an indicator that measures the level of importance regarding nutritional and health aspects through a questionnaire comprising eight items. This measurement uses a seven-point Likert scale, ranging from ‘not at all important’ to ‘extremely important’, to evaluate the importance placed on nutritional and health factors when selecting meals and food products. The eight dietary and nutritional aspects covered include the reduction of salt, reduction of sugar, reduction of artificial additives, reduction of saturated fats, reduction of calories, increase of vitamins, increase of dietary fiber, and increase of unsaturated fats. Cluster A is characterized by respondents with extreme response tendencies, indicating polarized views on the importance of nutrients; Cluster B consists of individuals who generally perceive all nutrients as unimportant, predominantly selecting the lower end of the scale to express their disregard for nutrient importance; Cluster C includes those who deem all nutrients as generally important, likely choosing higher scores on the scale to reflect their valuation of nutrient importance; and Cluster D comprises respondents who exhibit a neutral stance towards nutrient importance, frequently opting for the middle option on the scale, indicating an indifference or lack of conscious consideration towards the importance of nutrients.

## Data Availability

The datasets generated during and/or analysed during the current study are not publicly available due to ethical considerations but are available from the corresponding author on reasonable request.

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
