# Peer review of "Characterizing Healthy Dietary Practices in Japan: Insights from a 2024 Nationwide Survey and Cluster Analysis"

_nutrients, 2024, doi:10.3390/nu16101412_

Round 1

Reviewer 1 Report

Comments and Suggestions for Authors

Review comments on Nutrients-2983530 “Characterizing healthy dietary practices in Japan: insights from a 2024 nationwide survey and cluster analysis of 27,154 respondents.”

Utilizing UMAP and OPTICS methods, the researchers investigated dietary patterns and prioritization in food selection from a web-based survey conducted in the early of 2024 among approximately 28,000 Japanese adults. The results are interesting, which confirmed a fact that people usually consider it exists but there is scarce evidence to prove it, i.e., people in certain food selection clusters had different social demographic profiles, or if they are with different social demographic profiles many have different dietary patterns. The findings of this study may help to find different approaches to deal with the increase of lifestyle related chronic diseases with promotion of healthy dietary practices. Below are my comments for authors’ consideration for improvement:

1.       The survey was conducted in two weeks of early February 2024 with a 7-day food frequency questionnaire. Although many food items can be found yearly around, especially those frozen food items, however, there are still seasonal differences. Does this difference affect people’s responses to habitual dietary preference, food choices, and/or everyday meal experiences?  Please have a discussion.

2.       It may help readers easily to understand what the food groups and ranking food selections are surveyed if the two outcomes in 2.3 can be organized as tables. See example below.

Ten specific food groups consumed over past week (outcome 1)

Food group

Note

1.       Seafood

Fish, shellfish, fresh or proceed

……

10. milk

3.       Table 1 takes 4 pages; indeed, it contains very rich information, but it seems too much to bear. Some variables, eg, residences took huge space of the table, but it seems not be critical in identifications of the related clusters. Therefore, I would suggest one can simplify it using regions instead of cities in the table. If the city information was in fact used in clustering process, then, maybe you can list the details in a supplemental table. It can be done for other demographic variables such as education, income, occupation, comorbid, etc.

4.       Figure 1 shows the results of the UMAP method used in this study, but it may not help people’s understanding if without details of legends on what cluster 1 – 4 in outcome 1 and A – D in outcome 2 represent.  My suggestion would be either put it into appendix or describe them after table 2 & 3.

5.       Cluster 3 & 4 from outcome 1 and cluster C and D from outcome 2 are the main findings from this study, they are most of the participants in the analysis and certainly worth the most attention. However, the other two clusters (1 and 2 for outcome 1 and A and B for outcome 2) also take about 10% and 15%, respectively. It may not be appropriate to ignore them totally.

6.       There are some descriptions of similarity of cluster 3 and cluster D in discussion, but if there is a chi-square test using a contingency table for cluster 1 – 4 and cluster A -D, it may strength the argument that the two are highly associated.

7.       There are some typos or minor grammar issues. Page 5, line 165. “…,body-mass index (BMI) calculated from self-reported height and weight in kilograms” missing “height in meter”. Page 16, line 462 should it be “further” instead of “fourth”?

Comments on the Quality of English Language

minor issues as described above.

Author Response

Response to Reviewer #1

General comment:

Utilizing UMAP and OPTICS methods, the researchers investigated dietary patterns and prioritization in food selection from a web-based survey conducted in the early of 2024 among approximately 28,000 Japanese adults. The results are interesting, which confirmed a fact that people usually consider it exists but there is scarce evidence to prove it, i.e., people in certain food selection clusters had different social demographic profiles, or if they are with different social demographic profiles many have different dietary patterns. The findings of this study may help to find different approaches to deal with the increase of lifestyle related chronic diseases with promotion of healthy dietary practices. Below are my comments for authors’ consideration for improvement.

We are deeply grateful for your careful review and the valuable feedback you provided. Your guidance has been invaluable to us in improving our work.

Major Comments:

  1. The survey was conducted in two weeks of early February 2024 with a 7-day food frequency questionnaire. Although many food items can be found yearly around, especially those frozen food items, however, there are still seasonal differences. Does this difference affect people’s responses to habitual dietary preference, food choices, and/or everyday meal experiences? Please have a discussion.

Thank you for your insightful comment regarding the seasonal variations in food availability and their potential impact on our survey results. We have now included a discussion on how the timing of the survey in early February might have influenced the dietary responses due to seasonal food availability. This addition aids in addressing the limitations of our study and emphasizes the importance of considering seasonal variations when interpreting the results related to dietary preferences and food choices. The revised texts are as follows:

“Fourth, the timing of our survey could also affect the findings. It was conducted over two weeks in early February 2024, utilizing a 7-day food frequency questionnaire. While many food items are available year-round, particularly frozen foods, seasonal variations still exist and could influence respondents' habitual dietary preferences, food choices, and daily meal experiences. This potential seasonal bias is an important factor to consider in interpreting our results.” (Limitations, page 16, line 497)

  1. It may help readers easily to understand what the food groups and ranking food selections are surveyed if the two outcomes in 2.3 can be organized as tables. See example below. Ten specific food groups consumed over past week (outcome 1)

Thank you for your suggestion to clarify the presentation of the Dietary Variety Score (Outcome 1) in our manuscript. We have incorporated a detailed table (Table 1) that outlines the specific food groups and their definitions as you recommended. This modification will help readers better understand the components of the DVS and the scope of each food group. For Outcome 2, we have opted to retain the existing format, as it effectively communicates the use of a seven-point Likert scale to assess the importance of nutritional factors in food selection, and we believe this description is already clear and accessible. We appreciate your constructive feedback and believe that these changes significantly enhance the readability and comprehensibility of our findings. Thank you again for your valuable insights. The revised texts are as follows:

“The DVS is determined through responses to a questionnaire inquiring about the frequency of consumption of ten specific food groups over the preceding seven days. Respondents choose from four options for each food group: 'almost every day'; 'every other day'; '1 to 2 times'; or 'rarely eaten'. Scoring is binary, with 'almost every day' responses receiving one point and all other responses zero, thereby generating a binary variable for each food group. The food groups include seafood; soy and soy products; green and yellow vegetables; meats; eggs; fats and oils; seaweeds; tubers; fruits; and milk. Table 1 presents a detailed breakdown of these groups. The order of the food groups is randomized for each respondent, and answering all questions is mandatory.” (2.3. Outcomes, page 3, line 133)

  1. Table 1 takes 4 pages; indeed, it contains very rich information, but it seems too much to bear. Some variables, eg, residences took huge space of the table, but it seems not be critical in identifications of the related clusters. Therefore, I would suggest one can simplify it using regions instead of cities in the table. If the city information was in fact used in clustering process, then, maybe you can list the details in a supplemental table. It can be done for other demographic variables such as education, income, occupation, comorbid, etc.

Thank you for your feedback concerning the detailed nature of what was originally Table 1 (Table 2 in the revised manuscript). In response, we have simplified the table by grouping city-specific residences into broader regional categories, effectively reducing the complexity while retaining necessary information for analysis. We opted not to group other variables such as education, income, occupation, and comorbidities, to avoid the need for validating new groupings, which could introduce unnecessary complexity and potentially overload the readers with information. We have retained detailed demographic information in a supplemental table, where space constraints are less of a concern, ensuring that all pertinent data remains accessible for interested readers. However, we have updated the supplemental tables to clearly indicate which cities correspond to each region, ensuring transparency and ease of reference.

  1. Figure 1 shows the results of the UMAP method used in this study, but it may not help people’s understanding if without details of legends on what cluster 1 – 4 in outcome 1 and A – D in outcome 2 represent. My suggestion would be either put it into appendix or describe them after table 2 & 3.

Thank you for your constructive feedback concerning the presentation of Figure 1 in our manuscript. We have revised the figure legend to more clearly articulate the cluster details associated with Outcomes 1 and 2. We now explicitly mention that the detailed descriptions of Clusters 1-4 and Clusters A-D are located in the notes sections of Tables 2 and 3 (Tables 3 and 4 in the revised manuscript), respectively. This adjustment not only enhances clarity but also directly guides readers to the detailed cluster definitions. We appreciate your suggestions and believe that these changes significantly improve the comprehension and accessibility of our results. The revised texts are as follows:

“(a) Distribution of clusters detected by OPTICS on the two-dimensional reduced representation of the data; (b) UMAP clusters for two-dimensional reduced representation of the data annotated by the OPTICS generated clusters. Outcome 1 refers to the Dietary Variety Score (DVS), which assesses dietary diversity based on responses to a questionnaire with ten items. These items ask about the frequency of consumption of ten specific food groups—seafood, soy products, green and yellow vegetables, meats, eggs, fats and oils, seaweeds, tubers, fruits, and milk—over the past week. The scoring system is binary: daily consumption is awarded one point, while all other frequencies score zero, resulting in a binary score for each of the ten food groups. Clusters 1-4 in Outcome 1 are as described in Table 3. Outcome 2 refers to an indicator that measures the level of importance regarding nutritional and health aspects through a questionnaire comprising eight items. This measurement uses a seven-point Likert scale, ranging from 'not at all important' to 'extremely important', to evaluate the importance placed on nutritional and health factors when selecting meals and food products. The eight dietary and nutritional aspects covered include the reduction of salt, reduction of sugar, reduction of artificial additives, reduction of saturated fats, reduction of calories, increase of vitamins, increase of dietary fiber, and increase of unsaturated fats. Clusters A-D in Outcome 2 are as described in Table 4.” (Figure 1, page 10, line 268)

“Outcome 1 refers to the Dietary Variety Score (DVS), which assesses dietary diversity based on responses to a questionnaire with ten items. These items ask about the frequency of consumption of ten specific food groups—seafood, soy products, green and yellow vegetables, meats, eggs, fats and oils, seaweeds, tubers, fruits, and milk—over the past week. The scoring system is binary: daily consumption is awarded one point, while all other frequencies score zero, resulting in a binary score for each of the ten food groups. Cluster 1 includes individuals who consume meat or fats/oils daily; Cluster 2 consists of those who drink milk daily; Cluster 3 is characterized by respondents who maintain a well-rounded daily diet; and Cluster 4 includes individuals without a specific daily dietary pattern.” (Table 2, page 11, line 293)

“Outcome 2 refers to an indicator that measures the level of importance regarding nutritional and health aspects through a questionnaire comprising eight items. This measurement uses a seven-point Likert scale, ranging from 'not at all important' to 'extremely important', to evaluate the importance placed on nutritional and health factors when selecting meals and food products. The eight dietary and nutritional aspects covered include the reduction of salt, reduction of sugar, reduction of artificial additives, reduction of saturated fats, reduction of calories, increase of vitamins, increase of dietary fiber, and increase of unsaturated fats. Cluster A is characterized by respondents with extreme response tendencies, indicating polarized views on the importance of nutrients; Cluster B consists of individuals who generally perceive all nutrients as unimportant, predominantly selecting the lower end of the scale to express their disregard for nutrient importance; Cluster C includes those who deem all nutrients as generally important, likely choosing higher scores on the scale to reflect their valuation of nutrient importance; and Cluster D comprises respondents who exhibit a neutral stance towards nutrient importance, frequently opting for the middle option on the scale, indicating an indifference or lack of conscious consideration towards the importance of nutrients.” (Table 3, page 13, line 316)

  1. Cluster 3 & 4 from outcome 1 and cluster C and D from outcome 2 are the main findings from this study, they are most of the participants in the analysis and certainly worth the most attention. However, the other two clusters (1 and 2 for outcome 1 and A and B for outcome 2) also take about 10% and 15%, respectively. It may not be appropriate to ignore them totally.

The class consisting of individuals who consume meat or fats/oils daily (Cluster 1) showed tendencies that were partially similar to those of Cluster 4, while the cluster of daily milk drinkers (Cluster 2) was somewhat akin to Cluster 3. For instance, in terms of well-being and social isolation, Cluster 1 showed no significant difference from Cluster 4, whereas Cluster 2 differed significantly from Cluster 4. However, various dietary habits and experiences compared with Cluster 4 were generally scattered, and the number of significant variables was fewer in both Clusters 1 and 2 compared to Cluster 3. The interpretation of Cluster A, which showed extreme responses regarding the importance of nutrition, remains challenging, but for Cluster B, which generally perceives all nutrients as unimportant, dietary habits and experiences were relatively close to those of Cluster D, with no significant difference in well-being and social isolation compared to Cluster D. However, trends in dietary habits and experiences still varied among the clusters. Details can be found in Supplementary tables 2 and 3.

“Cluster 1, comprised of individuals who consume meat or fats/oils daily, exhibited tendencies that were partially similar to those observed in Cluster 4. Conversely, Cluster 2, characterized by daily milk consumption, showed similarities to Cluster 3. Regarding well-being and social isolation, Cluster 1 (17.3% and 69.6%, respectively) demonstrated no significant differences when compared to Cluster 4 (14.1% and 73.6%, respectively). In contrast, Cluster 2 (17.7% and 66.1%, respectively) exhibited significant differences from Cluster 4. Despite these findings, the dietary habits and experiences relative to Cluster 4 were generally varied, with both Clusters 1 and 2 showing fewer significant variables compared to Cluster 3. The interpretation of responses from Cluster A, which displayed extreme views on the importance of nutrition, continues to be challenging. On the other hand, Cluster B, which generally deems all nutrients as unimportant, had dietary habits and experiences that were closely aligned with those of Cluster D, with no significant differences in well-being and social isolation (16.8% and 72.3% vs 16.6% and 71.3%, respectively). Nonetheless, variations in dietary habits and experiences were still observed across the clusters. Further details are available in Supplementary Tables 2 and 3.” (3. Results, page 14, line 368)

  1. There are some descriptions of similarity of cluster 3 and cluster D in discussion, but if there is a chi-square test using a contingency table for cluster 1 – 4 and cluster A -D, it may strength the argument that the two are highly associated.

Thank you for your insightful suggestion to incorporate a chi-square test using a contingency table for clusters 1 – 4 and clusters A – D. We have added this analysis to our discussion, which indeed strengthens the argument for a significant association between these clusters. The results of the chi-squared test (p<0.001) are now clearly reported, underscoring the robustness of our findings. We appreciate your guidance which has enhanced the clarity and the statistical rigor of our study. The revised texts are as follows:

“When comparing Cluster C, which emphasizes the importance of diet, to Cluster D, which shows a lack of concern for dietary habits, similar patterns emerged (Supplementary Table 3). We performed a chi-squared test using a contingency table for clusters 1 – 4 and clusters A – D, revealing a significant association between these clusters (p<0.001). Members of Cluster C exhibited behaviors and preferences that mirrored those in Cluster 3, with 70.3% of Cluster 3's members belonging to Cluster C; similarly, 73.4% of Cluster C's members were also part of Cluster 3.” (3. Results, page 14, line 354)

  1. There are some typos or minor grammar issues. Page 5, line 165. “…,body-mass index (BMI) calculated from self-reported height and weight in kilograms” missing “height in meter”. Page 16, line 462 should it be “further” instead of “fourth”?

Thank you for your careful review and suggestions. We have made the corrections you pointed out, and have further revised the sequence indicator from "fourth" to "sixth" to accurately reflect the order of the points discussed in the manuscript. This correction ensures the logical progression and clarity of our discussion on the study's limitations.

Reviewer 2 Report

Comments and Suggestions for Authors

Shuhei Nomura et al. submitted to Nutrients an article, dealing with a nationwide survey and cluster analysis on the characterization of healthy dietary practices in Japan.

The main limitations of this study concern the self-reported responses, relating to a survey intended as an advertising campaign, useful for subsequently using the "bonuses" provided by the responses: a fairly eclectic method.

The title needs to be reshaped, not indicating the number of respondents.

It is necessary to detail the methods through which the survey was validated, also using a pilot study and indicating the Cronbach Alpha Coefficient.

The Authors should structure the findings emerging from this study according to a SWOT analysis logic.

Comments on the Quality of English Language

Minor editing of English language required

Author Response

Response to Reviewer #2

General comment:

Shuhei Nomura et al. submitted to Nutrients an article, dealing with a nationwide survey and cluster analysis on the characterization of healthy dietary practices in Japan.

Your thoughtful review and insightful feedback were invaluable. We are grateful for your guidance.

Major Comments:

  1. The main limitations of this study concern the self-reported responses, relating to a survey intended as an advertising campaign, useful for subsequently using the "bonuses" provided by the responses: a fairly eclectic method.

Thank you for your observations concerning our approach to managing biases in our study. We appreciate your positive recognition of how we used incentives to encourage participation and mitigate biases typically associated with self-reported data. In the revised manuscript, we have emphasized this strategy more clearly, noting its role in enhancing response rates from a broader participant base and addressing potential limitations. The revised texts are as follows:

“Second, self-selection bias may affect the representativeness of the survey participants. However, it is important to highlight that despite potential biases, the study effectively uses incentives in the form of points (usable for purchasing goods) by the survey company. These incentives are designed to motivate responses from individuals otherwise disinterested in the survey, thereby potentially mitigating some of the common limitations associated with self-reported data.” (Limitations, page 16, line 478)

  1. The title needs to be reshaped, not indicating the number of respondents.

Thank you for your valuable feedback regarding the title of our manuscript. We have revised the title to "Characterizing healthy dietary practices in Japan: insights from a 2024 nationwide survey and cluster analysis" in order to omit the specific number of respondents, following your suggestion.

  1. It is necessary to detail the methods through which the survey was validated, also using a pilot study and indicating the Cronbach Alpha Coefficient.

Thank you for your feedback regarding the validation of our survey methods. We acknowledge the importance of detailing our methodology; however, our study design did not include elements such as a pilot study or statistical reliability tests like the Cronbach's alpha coefficient. These elements require specific conditions, including repeated measures from the same respondents, which our survey did not involve. We have carefully developed our questionnaire based on validated questions from prior research and substantial expert input, consistent with established practices in our field. We have elaborated on these aspects in the methods and limitations sections to ensure clarity regarding the scope of our methodology. The revised texts are as follows:

“The questionnaire items were developed based on a thorough review of literature on similar topics, including studies conducted within the Japanese context [17-25]. This process was supervised by experts involved in the "Data for Social Transformation" academic platform, a collaboration among Japan's government, industry, academia, and social sectors. Although our methodology did not include elements such as a pilot study, the survey design integrated expert opinions and was reflective of methods validated in prior research.” (2.2. Measurement, page 3, line 116)

“Seventh, while the survey methodology employed in this study draws on expert opinions and a comprehensive literature review, it is important to note the limitations associated with the absence of a pilot study and repeated measurements from the same respondents. These conditions are typically required for certain statistical validations such as the calculation of reliability coefficients. Consequently, the interpretations of the survey results should be considered with these methodological constraints in mind.” (Limitations, page 17, line 513)

  1. The Authors should structure the findings emerging from this study according to a SWOT analysis logic.

We appreciate the reviewer's suggestion to structure our findings using a SWOT analysis; however, we believe that incorporating such a framework may not align closely with the specific objectives and methodological approach of our study. Our research is primarily focused on exploring dietary habits and their multifaceted impacts on health within a Japanese context, utilizing statistical techniques like dimensionality reduction and clustering to identify patterns and implications. The SWOT analysis, while valuable for strategic planning and management contexts, may not effectively capture the nuanced, empirical insights we aim to provide through our analytical methods. Thus, while we recognize the potential utility of a SWOT analysis in different research settings, we have chosen to maintain our current approach to more directly address our research questions and the evidence-based framework that underpins our study.

Round 2

Reviewer 2 Report

Comments and Suggestions for Authors

The Authors responded to the comments and suggestions in a more than satisfactory manner, contributing to increasing the robustness of the manuscript and providing clarifications of details that were indispensable. Thank you

Comments on the Quality of English Language

Minor editing of English language required